# Synergy in anti-malarial pre-erythrocytic and transmission-blocking antibodies is achieved by reducing parasite density

Ellie Sherrard-Smith[1†]*, Katarzyna A Sala[2†], Michael Betancourt[3], Leanna M Upton[2], Fiona Angrisano[2], Merribeth J Morin[4], Azra C Ghani[1‡], Thomas S Churcher[1‡], Andrew M Blagborough[2‡]

[1]MRC Centre for Global Infectious Disease Analysis, Department of Infectious Disease Epidemiology, Imperial College London, London, United Kingdom; [2]Department of Life Sciences, Imperial College London, London, United Kingdom; [3]Applied Statistics Center, Columbia University, New York, United States; [4]PATH's Malaria Vaccine Initiative, Washington, United States

**Abstract** Anti-malarial pre-erythrocytic vaccines (PEV) target transmission by inhibiting human infection but are currently partially protective. It has been posited, but never demonstrated, that co-administering transmission-blocking vaccines (TBV) would enhance malaria control. We hypothesized a mechanism that TBV could reduce parasite density in the mosquito salivary glands, thereby enhancing PEV efficacy. This was tested using a multigenerational population assay, passaging *Plasmodium berghei* to *Anopheles stephensi* mosquitoes. A combined efficacy of 90.8% (86.7–94.2%) was observed in the PEV +TBV antibody group, higher than the estimated efficacy of 83.3% (95% CrI 79.1–87.0%) if the two antibodies acted independently. Higher PEV efficacy at lower mosquito parasite loads was observed, comprising the first direct evidence that co-administering anti-sporozoite and anti-transmission interventions act synergistically, enhancing PEV efficacy across a range of TBV doses and transmission intensities. Combining partially effective vaccines of differing anti-parasitic classes is a pragmatic, powerful way to accelerate malaria elimination efforts.

DOI: https://doi.org/10.7554/eLife.35213.001

*For correspondence:
e.sherrard-smith@imperial.ac.uk

†These authors contributed equally to this work
‡These authors also contributed equally to this work

**Competing interests:** The authors declare that no competing interests exist.

## Introduction

Malaria remains a major global health challenge with an estimated 216 million new cases and 445,000 deaths in 2016 (*World Health Organization, 2017*). Whilst current tools have substantially reduced the global burden of disease, new tools will be needed to achieve malaria elimination (*Walker et al., 2016*). Early development of malaria vaccines focused on either the pre-erythrocytic stage vaccine (PEV) – eliciting an immune response to prevent incoming sporozoites from establishing patent infection – or blood-stage – boosting natural responses to surface proteins on the infected erythrocytes (*Schwartz et al., 2012*). The first malaria vaccine RTS,S/AS01 to complete Phase III trials is a PEV vaccine and has been demonstrated to be partially effective, reducing clinical incidence in 5 – 17-month-old children by 36.3% (95%CI: 31.8 – 40.5%) over 40 months follow-up (*RTSS Clinical Trials Partnership, 2015*). Further candidate PEV vaccines include those that achieve protective immunity through irradiated/chemo-attenuated *Plasmodium falciparum* sporozoites, (e.g. PfSPZ vaccines [*Seder et al., 2013*]), those that use viral vectors to induce T-cell responses to provide protection (*de Barra et al., 2014*; *MVVC group et al., 2015*) and the promising next-generation RTS,S-like vaccine, R21 (*Collins et al., 2017*). A range of vaccines that target human-to-mosquito transmission by attacking sexual, sporogonic, and/or mosquito antigens

**eLife digest** In 2016, malaria caused an estimated 216 million illnesses and 445,000 confirmed deaths globally. The disease is caused by a parasite, and mosquitos infected with the parasite transmit them to humans when they bite. In humans, the parasites enter the body and head to the liver before spending part of their life cycle in red blood cells, which cause the symptoms of the disease.

Prevention efforts have reduced the burden of malaria but eliminating the disease will require new tools. One option is to use vaccines. The world's first malaria vaccine – a so-called pre-erythrocytic vaccine (PEV) – targets the stages preceding the parasite reaching the liver. This vaccine prevents malaria parasites from infecting people, but it is only partially effective. Scientists are also developing transmission-blocking vaccines (TBVs). These TBVs block the development of malaria parasites in mosquitos that bite vaccinated humans. So far, the most promising TBV candidates are also only partly effective.

It is possible that using PEV and TBV vaccines together could boost their effectiveness, since the TBV vaccines reduce the number of parasites that infect each mosquito. This means that fewer parasites are injected into the next person. Currently, the PEVs work better when there are fewer parasites infecting a person.

Now, Sherrard-Smith et al. show that combining TBVs with PEVs enhances their antimalarial effects. In the experiments, Sherrard-Smith et al. treated mice with either TBV or PEV vaccines, or both. Then, the mice were exposed to mosquitos infected with the malaria parasite. As expected, the TBV and PEV treatments were only partially effective when used alone. But exposing the mice to both TBVs and PEVs eliminated the parasites from the mosquitos and the mice.

The combined benefit of TBVs and PEVs were greater than would be expected if either vaccine was acting alone and the effects were simply multiplied, suggesting they enhance each other's effects. More studies of TBVs in humans are needed to prove they are safe and effective in the real world. More studies also are needed to confirm what Sherrard-Smith et al. found in mice would happen in humans treated with a combination of TBV and PEV vaccines. But if such future studies prove this combination approach is effective, it could be a powerful tool in the fight against malaria.
DOI: https://doi.org/10.7554/eLife.35213.002

(transmission-blocking vaccines, TBV) are also under development (*Talaat et al., 2016*; *Wu et al., 2008*). Pre-clinical investigations have identified multiple antigens (e.g. Pfs25, P230, P48/45) as targets for TBV candidates that, when administered, can reduce transmission to mosquitoes (*Hoffman et al., 2015*; *Sauerwein and Richie, 2015*), but complete, or reproducible, translation to the clinic has not been achieved so far (*Talaat et al., 2016*; *Hoffman et al., 2015*; *Sauerwein and Richie, 2015*).

One of the major challenges encountered in developing PEV malaria vaccines is the partial protection achieved against each exposure, despite high levels of induced antibody titres. It has been hypothesized that this may be in part due to the over-dispersed distribution of sporozoites in each infectious bite, such that despite inducing a high per-parasite killing efficacy, the probability that at least one parasite reaches the liver and progresses to blood-stage infection remains high (*White et al., 2013*; *Bejon et al., 2005*). The classical approach to overcoming this is to attempt to further increase either the quantity, breadth or quality of the immune response (*Remarque et al., 2012*; *Courtin et al., 2009*; *Chaudhury et al., 2016*). We hypothesized that an alternative mechanism would be to combine a PEV with approaches that reduce the number of sporozoites in the mosquito salivary glands. TBVs have been demonstrated to act in this way, reducing ookinete and sporozoite density (*Bompard et al., 2017*; *Blagborough et al., 2013*). We sought therefore to identify whether this mechanism could result in synergistic interactions between PEVs and TBVs co-administered within a population.

To test this hypothesis, we used an established murine population assay to investigate the clearance of malaria over multiple generations in a controlled laboratory environment (*Blagborough et al., 2013*). Here, the rodent malaria parasite *Plasmodium berghei* is passed between populations of mice by the direct feeding of *Anopheles stephensi* mosquitoes. To simulate

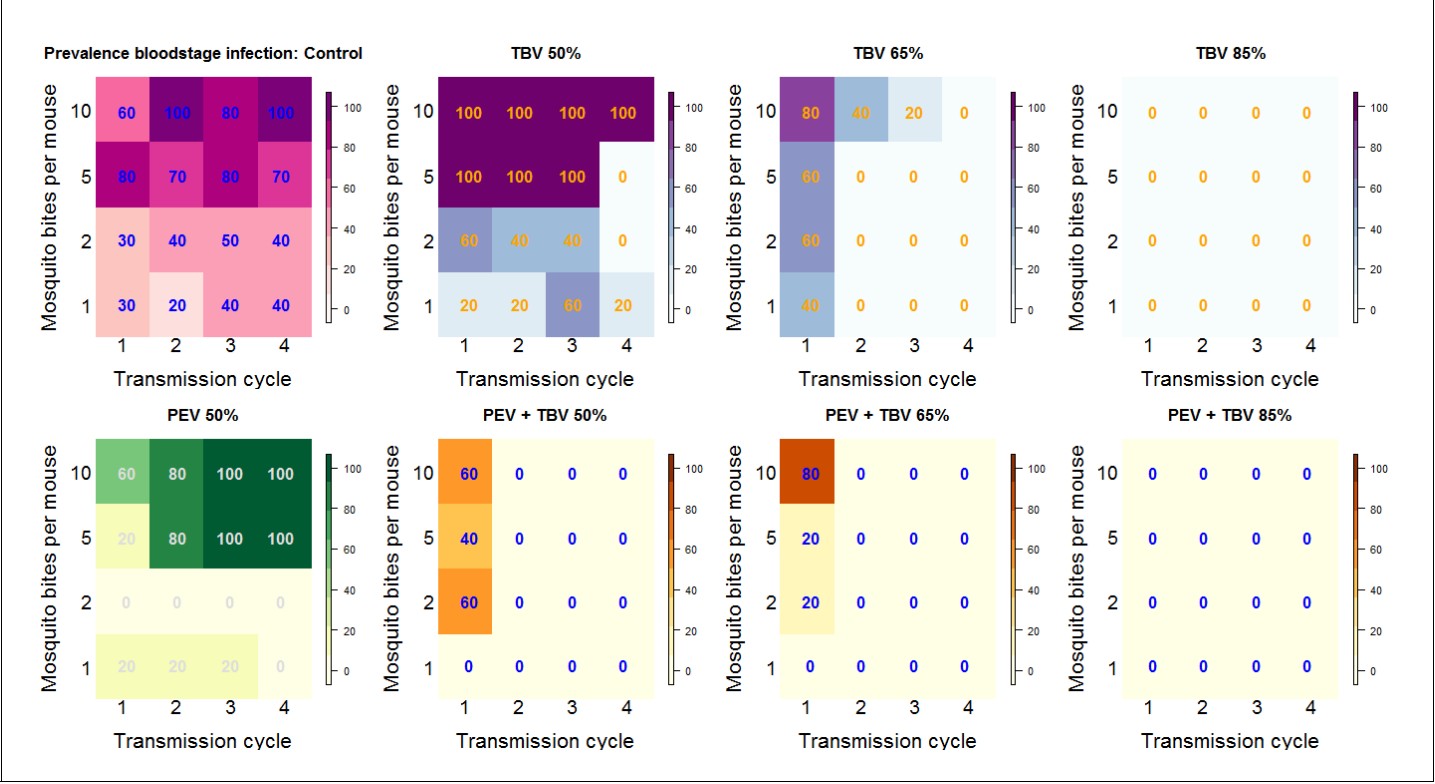

**Figure 1.** Summary outcome data. The number in each box (and the color) shows the percentage of mice infected for each treatment arm of the experiment by transmission cycle and biting rate (the number of potentially infectious mosquito bites received per mouse). In transmission cycle 0, all mice are infected (not shown), where a TBV antibody is administered (top row: purple and blue) infections in mice are progressively reduced. The antibody works best at the higher dose (TBV 85%) and lower biting rates. The PEV antibody (lower row: yellow and green) is effective at lower biting rates (where each mouse received one or two potentially infectious mosquito bites per transmission cycle) but is not able to reduce infection at higher biting rates. In combination, at any biting rate, the combined antibodies (lower row: orange) always cleared infections by transmission cycle 2.
DOI: https://doi.org/10.7554/eLife.35213.003

The following figure supplement is available for figure 1:

**Figure supplement 1.** Adapted from Malaria Journal (*Sherrard-Smith et al., 2017*) A graphical outline of the multi-generational transmission experiment (**A**) and its mathematical representation (**B**).
DOI: https://doi.org/10.7554/eLife.35213.004

the antibody response to a PEV, a monoclonal antibody (mAb-3D11) - which targets the same corresponding parasite circumsporozoite protein (CSP) as RTS,S – was passively transferred (intravenously) into mice. To act as a partially effective PEV (broadly comparable to RTS,S), a mAb-3D11 dose was selected which reduces mosquito-to-mouse transmission probability by ~50% (as evaluated in naive mouse populations, transfused with differing doses of 3D11, challenged with five mosquito bites, see Materials and methods). Sterilizing immunity of 47.2% was titrated over multiple challenges. The complementary actions of a TBV antibody response were simulated using an anti-Pfs25 monoclonal antibody 4B7 (mAb-4B7), administered by passive transfer, which targets the same parasite stages as the most currently advanced human TBV candidate, Pfs25 (*Talaat et al., 2016*). This well-established transmission-blocking monoclonal antibody was used in combination with a transgenic *P. berghei* (*Pb*Pfs25DR3) parasite that expresses Pfs25 in place of its rodent homologue. *Pb*Pfs25DR3 is phenotypically indistinguishable to WT *P. berghei*, expresses Pfs25 on the surface of the zygote/ookinete, and has been used previously to assay a range of TBVs (*Goodman et al., 2011*; *Kapulu et al., 2015*). A series of transfused mAb-4B7 doses were tested in multiple (n = 6) direct feeding assays to titrate the appropriate doses to generate a 50%, 65% and 85% reduction in transmission to the mosquito (measured as reduction in oocyst prevalence) (see Materials and methods).

**Table 1.** Summary of the density model estimation of efficacy against prevalence and parasite density for the transmission blocking (TBV) and pre-erythrocytic (PEV) antibodies that were administered either alone or together to reduce malaria parasites in mice. The interaction between the two antibodies are measured using the ratio of observed estimates for combination treatments compared to the expected efficacy were vaccine antibodies acting independently using the simulated posterior draws from the density model (*Sherrard-Smith et al., 2017*). A value of less than one indicates an antagonistic interaction, = 1 suggests antibodies are acting independently, and greater than one shows synergy.

| Intervention arm | Efficacy | | Synergy | | | |
|---|---|---|---|---|---|---|
| | Reduction in prevalence (95% credible intervals) | Reduction in density (95% credible intervals) | Prevalence | p-value | Density | p-value |
| Individual vaccine efficacies | | | | | | |
| All TBV combined | 68.0 (61.1–74.1) | 51.5 (6.8–72.9) | | | | |
| TBV: MAb-4B7 (50%) | 33.9 (18.2–47.4) | 13.6 (0–47.5) | | | | |
| TBV: MAb-4B7 (65%) | 74.3 (65.7–82.4) | 69.3 (47.8–84.8) | | | | |
| TBV: MAb-4B7 (85%) | 95.8 (90.2–100) | 94.2 (79.1–100) | | | | |
| PEV: Mab-3D11 (50%) | 48.0 (36.6–58.0) | 2.8 (0–25.2) | | | | |
| Combined vaccine efficacies | | | | | | |
| PEV and All TBV combined | 90.8 (86.9–94.2) | 90.9 (79.0–96.4) | 1.09 (1.02–1.18) | p<0.0035 | 2.08 (1.20–5.02) | p<0.0025 |
| PEV (50%) and MAb-4B7 (50%) | 82.2 (74.6–88.9) | 85.8 (75.6–92.8) | 1.27 (1.07–1.60) | p<0.0015 | 19.04 (1.56–75.16) | p<0.0001 |
| PEV (50%) and MAb-4B7 (65%) | 92.8 (86.1 - 98.6) | 93.2 (75.4 - 99.7) | 1.07 (0.98 - 1.17) | p<0.0675 | 1.36 (1.02 - 1.97) | p<0.02 |
| PEV (50%) and MAb-4B7 (85%) | 96.9 (91.2 – 100) | 94.8 (74.4 - 100) | 0.99 (0.93 - 1.04) | p<0.5435 | 1.01 (0.78 - 1.22) | p<0.3755 |

DOI: https://doi.org/10.7554/eLife.35213.005

The following source data available for Table 1:

**Source data 1.** The raw data used for analysis.

DOI: https://doi.org/10.7554/eLife.35213.006

We undertook a series of experiments with either PEV alone, TBV alone at three different efficacies (50%, 65% and 85%), or combinations of the two across four generations of transmission. For each experiment mice were exposed to either 1, 2, 5 or 10 infectious mosquito bites to allow estimation of combined transmission-blocking efficacies between 20% and 100% (Materials and methods).

## Results

At lower TBV antibody dose exposure levels (50% and 65%), the probability of eliminating all parasites from the mouse/mosquito populations was greater when PEV and TBV antibodies were administered together compared to singly, irrespective of the dose of the TBV antibody administered (*Figure 1*). Using statistical methods that explicitly capture parasite density and account for the impact of the interventions on both the prevalence and density of infection (*Sherrard-Smith et al., 2017*), we estimated PEV antibody alone to reduce the prevalence of infection by 48.0% (95% credible interval, CrI, 36.6–58.0%). Similarly, use of TBV antibody alone reduced the prevalence of infection by 33.9% (95% CrI: 18.2–47.4%), 74.3% (65.7%–82.4%) and 95.8% (90.2%–100%) for the 50%, 65% and 85% individual efficacy titres, respectively. If the actions of the two antibody types were to act independently, the predicted combined efficacy would be 83.3% (79.1–87.0%). A substantially higher efficacy of 90.8% (86.7–94.2%) (p=0.0035) was observed in the PEV +TBV antibody group, indicating a synergistic interaction (*Table 1*). Here, the 95% Credible Intervals did not overlap the median estimates, demonstrating a significant difference between the two treatments. The same relationship was observed when examining vaccine efficacy against parasite density (p=0.0025) (*Table 1*).

Sub-dividing the data by TBV antibody dose, the greatest synergistic enhancement in efficacy against parasite prevalence was seen at lower doses of functional TBV antibodies (*Figure 2*). A TBV

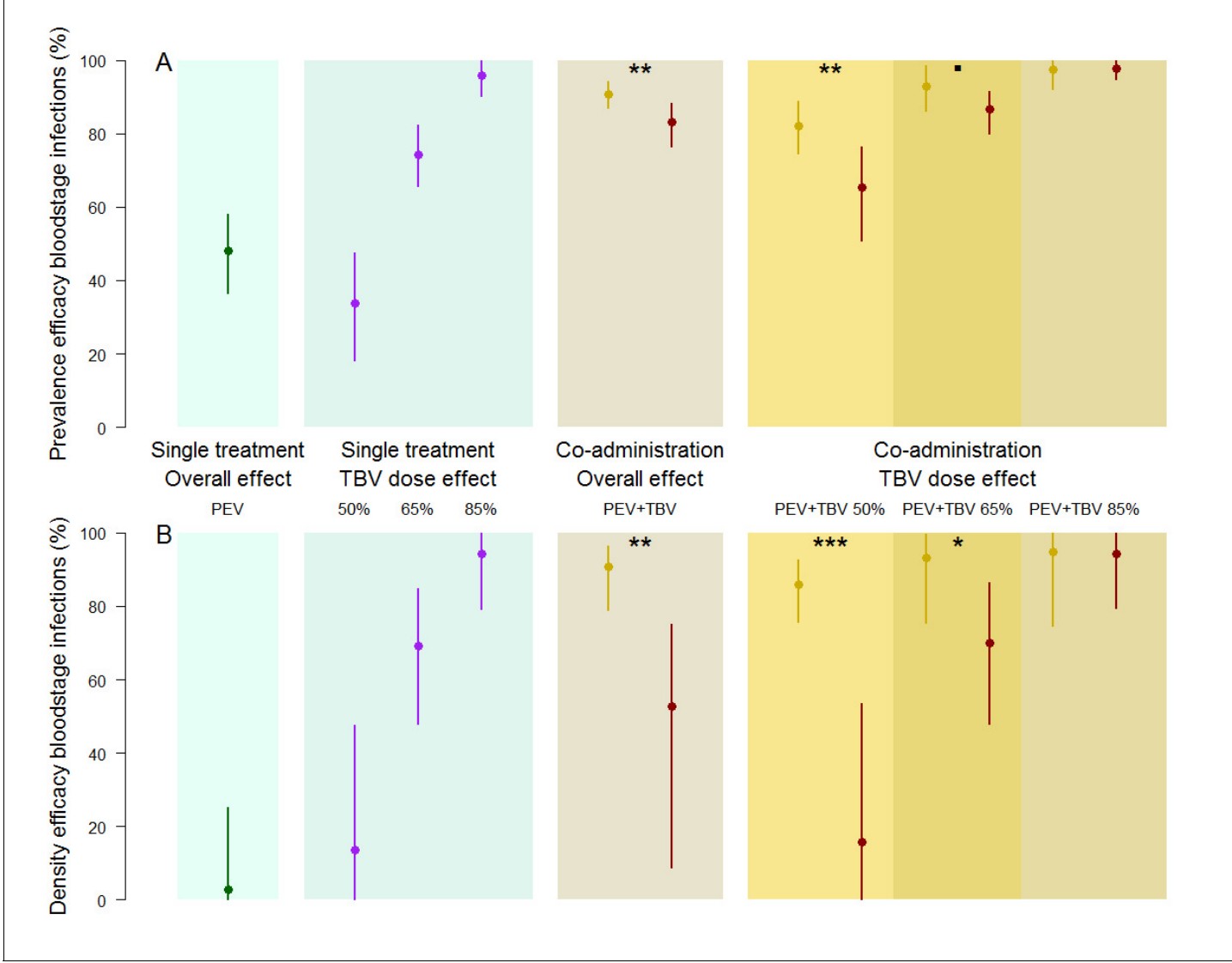

**Figure 2.** The efficacy of vaccine antibody combinations against parasite prevalence (the proportion of infected hosts) (**A**) and density (the mean parasite density per host, measured as the number of infected red blood cells out of a total subsample of 1200 erythrocytes) (**B**) in mice. The observed efficacy of PEV monoclonal antibody mAb-3D11 at a dose previously shown to reduce transmission to mice by ~50% when exposed to five infectious mosquito bites. The efficacy of the TBV mAb-4B7 at doses previously shown to reduce transmission to mosquitoes by 50%, 65% and 85% (blue section, purple lines). The efficacy of antibodies administered together (gold sections) that were observed (gold) or expected (red) were efficacies for each antibody acting independently (mean (point) and 95% credible intervals (lines) shown). Asterisks indicate increasing levels of support of synergy (p<0.1•, p<0.05*, p<0.01**, p<0.001***).
DOI: https://doi.org/10.7554/eLife.35213.007

antibody dose which reduces mouse-to-mosquito transmission by 50% increased the efficacy of PEV antibody prevalence to 82.2% (74.6–88.9%) compared to an expected 65% (57.7–73.2%) efficacy if the vaccines acted independently, a strong indication of synergy (p=0.0015). Similar synergistic effects against parasite density were observed (*Figure 2*, p <0.0001). Weaker synergistic effects were observed against parasite prevalence and density for the PEV +TBV antibody at the 65% dose (*Figure 2*, p=0.0675, 0.02, respectively) (*Table 1*). At the highest TBV antibody dose (85%), the TBV alone already reduced parasites in the population to low levels, hence there was insufficient power to detect further synergy between the interventions as all parasites were eliminated from the experimental population (*Figure 1*).

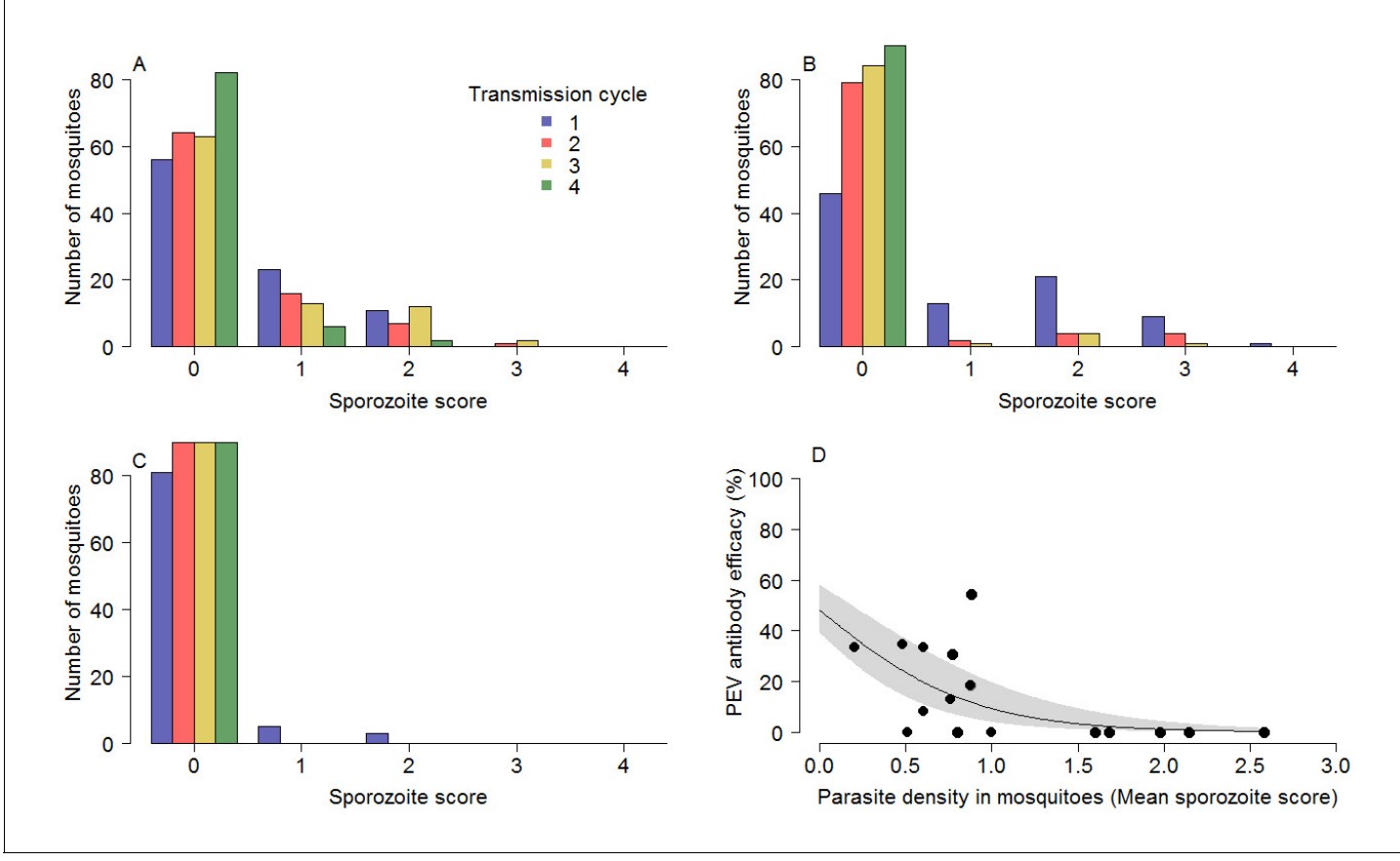

**Figure 3.** The impact of TBV monoclonal antibody mAb-4b7 on PEV mAb-3D11 efficacy. The TBV antibody alone reduces the tail of the sporozoite distribution in mosquitoes across transmission cycles for the (**A**) 50% dose, (**B**) 65% dose and (**C**) 85% dose. (**A–C**) demonstrate that the sporozoite scores (measured on a log scale; 0 indicates no sporozoites; 1 = 1–10 sporozoites; 2 = 11–100 sporozoites; 3 = 101–1000 sporozoite and; 4 =>1000 sporozoites per mosquito) tend toward zero for successive transmission rounds. (**D**) The efficacy of the PEV antibodies to prevent mosquito-to-mouse-to-mosquito transmission (a percentage reduction in sporozoite infections in mosquitoes) is greatest at lower mosquito parasite densities (as assessed by sporozoite score following blood-feeding).
DOI: https://doi.org/10.7554/eLife.35213.008

The impact of circulating TBV antibody on the reduction of parasites within the mosquito explains the greater efficacy of the PEV antibody in the combination treatment groups. The presence of TBV antibody (mAb-4B7) reduced oocyst counts in infected mosquitoes (ANOVA: $F_{1,1525}$ = 75.3, p<0.0001). Similarly, TBV antibody presence reduced sporozoite density in infected mosquitoes (ANOVA: $F_{1,707}$ = 163.9, p<0.0001). *Figure 3A,B and C* further illustrate the effect that TBVs have on sporozoite density distribution for the TBV-50%, 65% and 85% single treatment groups respectively. At each dose, the tail of the distribution of sporozoites is curtailed progressively across transmission cycles. *Figure 3D* illustrates that the efficacy of the PEV is density-dependent, with higher efficacy achieved when mosquitoes have lower sporozoite density.

## Discussion

The efficacy of multiple PEV and TBV candidates against rodent and human parasites have been shown to depend on parasite density; TBV (anti-Pfs25, anti-Pfs48/45 and immune blocking serum) efficacy decreases with increased parasite dose (*Churcher et al., 2012*; *Miura et al., 2016*) and a representative PEV (RTS,S) only provides sterilizing immunity in volunteers against lightly infected mosquitoes (*Churcher et al., 2017*). This suggests that both types of vaccine can halt transmission against a defined (but currently uncharacterized) quantity of parasites, which is enough to prevent

onward infection in lightly infected mosquitoes/humans, but that onward transmission is still possible from heavily infected mosquitoes or individuals.

Here, a partially effective TBV antibody reduced the number of parasites in infected mosquitoes, ensuring that the PEV antibodies encounter fewer parasites than would otherwise be expected if only a single antibody/vaccine class was administered in isolation. Thus, the subsequently reduced parasite burden increased the efficacy of the PEV when co-administered with a TBV. Potentially, a synergistic response could be induced by an unspecified biochemical or immunological interaction between the two vaccines. This explanation can be discounted within this system as the experimental design results in mosquito-to-mouse transmission being measured prior to the administration of a transmission-blocking intervention (passive transfer of mAb-4B7 1 hr prior to blood feed) as a new, naive batch of mice were infected in each generation.

The greatest synergy was observed in the lower TBV dose group, although it is likely to operate across all TBV and PEV doses when vaccine reduces parasite density but fails to clear infection from host or vector. The 85% TBV dose administered alone eliminated the parasite over a single generation without the action of the PEV so there was no opportunity to show synergy. A 100% efficacious PEV or TBV would not require augmenting with an alternative vaccine, although their efficacy is still likely to drop over time as antibodies decay so combining highly effective vaccine may still be advantageous depending on the relative rates of antibody loss.

While the direct translatability of rodent experiments to human health is variable, this approach is invaluable to demonstrate unequivocally that the mechanism behind the observed synergy is the direct result of TBV antibody reducing parasite density in infected mice and thereby enabling the density-dependent PEV antibody to be optimal (*Figure 3*). This mechanism is very likely to mirror that for humans, which cannot be tested directly due to ethical considerations and complex environmental variation. Whilst the murine system uses *P. berghei,* the mechanism of action of the PEV antibodies administered is matched to the antibody-based mechanism of the RTS,S vaccine; that is sporozoite invasion of the liver is inhibited by the presence of CSP-targeted antibodies in mice both in vivo and ex vivo (*Grüner et al., 2003*). The *P. berghei* strain used here is genetically modified to express the human TBV candidate, *P. falciparum* P25 (Pfs25) in place of its *P. berghei* counterpart. Thus, a proven anti-falciparum TBV mAb (4B7) can be used directly within the model (*Goodman et al., 2011*). The evidence that co-administering TBV and PEV antibodies can accelerate toward controlling malaria transmission is a first step toward trialing such combinations in more natural parasite-vector-host combinations and environments. There is good reason to believe that the population dynamics of parasites and partially effective vaccines may be similar in human malaria. Transmission of human malaria from human-to-mosquito and mosquito-to-human is also considered to depend on the density of the parasite (*Churcher et al., 2013*; *Sinden et al., 2007*). The average number of oocysts in wild caught mosquitoes is likely to be substantially lower than the numbers observed in the rodent system (*Rosenberg, 2008*) but oocyst distribution is highly over-dispersed (*Medley et al., 1993*) meaning that some mosquitoes have very-high-density infections. These highly infected mosquitoes are likely to be more infectious, so reducing their frequency through adding a TBV, could have additional impact on overall transmission. The epidemiological importance of any synergistic interaction between vaccine types in the field is hard to predict and will depend on many confounding factors such as human immunity, drug treatment, vector susceptibility, antigen escape, amongst others. Transmission is likely to be highly heterogeneous, caused by factors such as vaccine non-responders and super-transmitting hosts. The impact of different types of heterogeneity can be investigated under controlled laboratory scenarios using the murine population assay, varying the vaccinated coverage, antibody dose and changing biting heterogeneity within a population. This could help understand the relative importance of these different heterogeneities and could be used to support the design of appropriately powered Phase III trials (or alternative trial designs) to fully assess the impact of combining vaccine components with alternative mechanisms of action.

There is no 'magic bullet' intervention against malaria and the current global strategy is to combine vector control and drug treatment tools in a timely manner to move towards malaria elimination. Our results suggest that the same approach might be taken for the use of vaccines and comprises the first practical demonstration that combining TBVs and PEVs may have auxiliary benefits. Synergism between PEV and TBVs could potentially enhance the efficacy of the current PEV vaccines, resulting in reduced burden and potentially elimination in areas where it was not previously possible. The development of novel anti-malarial vaccines is both costly and time-consuming.

Combining partially effective vaccines of differing anti-parasitic classes may be therefore a pragmatic and powerful way to accelerate malaria elimination efforts. Synergism between PEV, TBVs and potentially a blood-stage vaccine (either administered separately or as a multi-component vaccine) could potentially enhance the efficacy of single vaccines, resulting in reduced burden and potentially elimination in areas where it was not previously possible.

# Materials and methods

## Key resources table

| Resource type (species) or resource | Designation | Reference | Identifiers |
|---|---|---|---|
| Antibody | mAb-4b7 | (*Stura et al., 1994*) | RRID:AB_2728658 |
| Antibody | mAb-3D11 | (*Mishra et al., 2012*) | RRID:AB_2728657 |

## Transmission-blocking vaccine surrogate: monoclonal antibody 4B7 (mAb-4B7)

The TBV mAb-4B7 neutralises the protein Pfs25 in sexual stages of the human malaria *P. falciparum* and reduces transmission of the parasite from host-to-mosquito (*Stura et al., 1994*). The transgenic murine malaria parasite, *P. berghei Pb*Pfs25DR3, expressing native Pfs25 in place of its rodent homologue, was used so that the same TBV antibody candidate could be examined within a mouse model (*Goodman et al., 2011*). MAb-4B7 was administered and examined at sub-optimal concentrations titrated to reduce oocyst prevalence in the mosquito midgut (as assessed using a direct feeding assay) by either 50, 65 or 85%. Given the severity of malaria, the WHO, the Strategic Advisory Group of Experts (SAGE) on Immunization and the Malaria Policy Advisory Committee (MPAC) recommended the RTS,S vaccine could be implemented in pilot countries in October 2015 (http://www.malariavaccine.org/malaria-and-vaccines/first-generation-vaccine/rtss, accessed 04/04/2018) when the vaccine was demonstrating relatively low efficacies of just 36.3% in children aged 5 to 17 months (*RTSS Clinical Trials Partnership, 2015*). These TBV doses were chosen to bridge a range of malaria vaccine efficacies that might be acceptable by WHO, SAGE and MPAC. Briefly, to titrate the appropriate dose, female Tuck Ordinary (TO) mice (6–8 weeks old, Harlan, UK) were treated with phenylhydrazine, and three days later, infected with $10^6$ *P. berghei Pb*Pfs25DR3 (*Goodman et al., 2011*). Three days later, infected mice were injected intravenously (*i.v.*) with 200 µl of purified mAb-4B7 at a range of doses. Negative control mice were transfused with 200 µl of phosphate buffered saline (PBS). After 1 hr, mice were anesthetised and 50 *Anopheles stephensi* mosquitoes (line SD 500, previously starved for 24 hr) were allowed to feed on each individual mouse. Mosquitoes were maintained as described in (*Blagborough et al., 2013*), and after 10 days, 50 mosquitoes were dissected and microscopically examined to measure oocyst intensity and prevalence. This was repeated five times, with *i.v.* administered doses of mAb-4B7 ranging from 0 µg to 750 µg. Prevalence efficacy was estimated as a function of mAb-4B7 concentration using a generalised linear model framework (*Bolker et al., 2009*) in which experimental replicate was treated as a random effect. A Gompertz function (*Churcher et al., 2013*) was fitted to the data using maximum likelihood methods. Mean concentrations were estimated using the best-fitting model (determined by log-likelihood tests) with 95% confidence intervals obtained from the profile likelihood. We estimated that a mAb-4B7 dose of 284.2 µg *i.v* (244.7–337.3 µg) was required to achieve a reduction in prevalence of 50%, a dose of 371.8 µg *i.v* (319.8–442.8 µg) for 65% reduction and a dose of 629.5 µg *i.v* (525–777.1 µg) for an 85% reduction. These calculated doses were then used in the mosquito-mouse model system as described below.

## Pre-erythrocytic vaccine surrogate: monoclonal antibody 3D11 (mAb-3D11)

The anti-*P. berghei* CSP mAb-3D11 (*Mishra et al., 2012*) is mechanistically similar to the recently registered RTS,S vaccine for human malaria, in that the presence of CSP-targeted antibodies in mice inhibit sporozoite invasion of the liver both in vivo and ex vivo (*Grüner et al., 2003*). An appropriate dose for mAb-3D11 was estimated from 40 individual passive transfers, administering a range of mAb-3D11 doses (0–150 µg i.v.) to mice (5 to 10 mice per experiment) and determining the

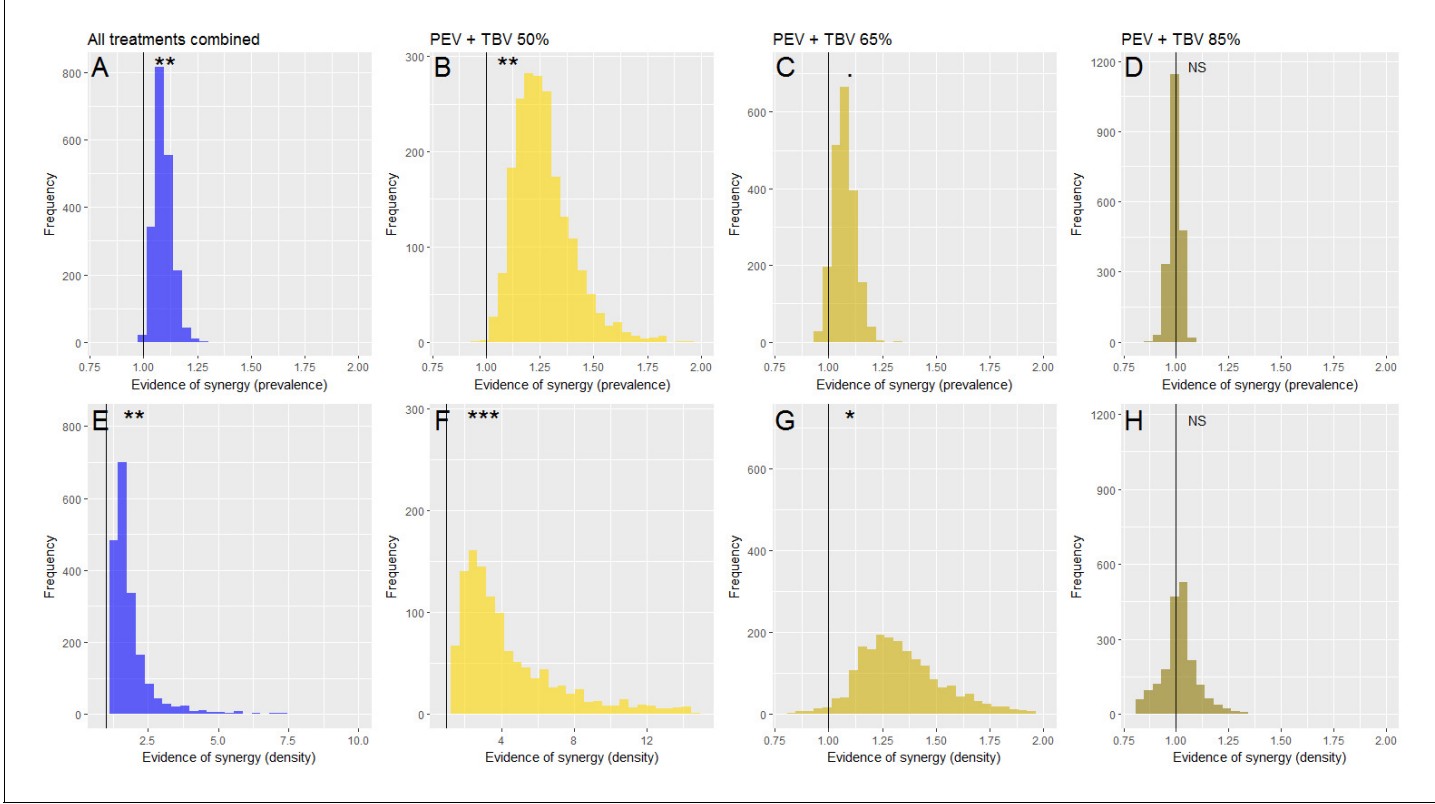

**Figure 4.** Evidence of a synergistic effect of parasite prevalence (**A–D**) or parasite density (**E–H**) of combining TBV monoclonal antibody (mAb-4B7) with PEV mAb-3D11. The frequency histograms show the probability that antibodies administered together have a higher efficacy than if they acted independently. In all panels, the vertical black line highlights the value 1, above this line indicates synergy, below it denotes an antagonistic interaction and falling at one indicates that the antibodies are acting independently (evidence for synergy p<0.1., p<0.05*, p<0.01**, p<0.001***). Taking all data together (**A** and **E**), there is a synergistic effect of combining vaccine antibodies as the majority of iterations fall above 1. This effect is stronger at lower doses of mAb-4B7 (**B**, **C**, **F** and **G**, note the different x-axes values indicating stronger support for lower TBV antibody doses), and stronger against parasite density (**E–H**) than parasite prevalence.

DOI: https://doi.org/10.7554/eLife.35213.009

prevalence efficacy at the given dose. A logistic function was fitted to these data using RStan (*Stan Development Team, 2017*) to determine the mean dose and 95% credible intervals that produces a ~50% reduction in the probability of infection. Consequently, mAb-3D11 antibody was administered at a single sub-optimal dose (50 µg i.v.) that prevented 47.2% (38.0–62.0% 95%CI) of transmission to mice that each received 5 potentially infectious mosquito bites. This dose was selected to match the approximate observed protection by RTS,S in human clinical trials.

## The mosquito-mouse model system

The mouse-to-mouse transmission model has been described in detail previously (*Upton et al., 2015*; *Blagborough et al., 2013*) (*Figure 1—figure supplement 1*). Briefly, five female (TO) mice (6–8 weeks old, Harlan, UK) were treated with phenylhydrazine, and, 3 days later, were infected with $10^6$ *P. berghei* *Pb*Pfs25DR3 (*Goodman et al., 2011*). Three days later, groups of infected mice were treated with the TBV antibody. After 1 hr, the mice were anesthetised and 500 *An. stephensi* mosquitoes (line SD 500, starved for 24 hr) fed randomly on the five infected mice within each group. Mosquitoes were maintained as described in *Blagborough et al., 2013*. After 10 days, a sub-sample of 50 mosquitoes were microscopically examined to measure oocyst intensity and prevalence. After 21 days post-feeding, sporozoites are present in the salivary glands and are maximally infectious to the vertebrate host (*Blagborough et al., 2013*). At this point, pre-defined numbers of mosquitoes (to simulate mosquito biting rates of 1, 2, 5 and 10 mosquito bites per mouse) were then randomly selected from the remaining mosquitoes and fed, for 20 min, on anesthetized mice from a naïve

cohort. The mosquito biting rate is an aspect of the experimental design that can be varied to be able to estimate the effect size more precisely. If the mosquito biting rate is small (say 1 or 2), the probability that the infection is eliminated rapidly in the intervention arm of the experiment is high (>80%) for TBD/TBV with efficacies above 40%. Thus, we cannot discriminate at a low mosquito biting rate between a TBD/TBV with 60% efficacy and one with 80% efficacy (they both eliminate). However, we do obtain a high degree of discrimination between an efficacy of 20% and one of 40%. Thus, a small mosquito biting rate is needed to get a precise estimate of the effect size of a TBD/TBV with lower efficacy. The converse also holds, so that a high mosquito biting rate (up to around 10 based on our initial experiment) is needed to obtain a precise estimate of a TBV/TBD with >80% efficacy. Using multiple mosquito biting rates increases the overall precision of our estimate of prevalence efficacy.

Each group of mice (five mice per group) either received the PEV antibody or no -intervention (negative control). Engorged mosquitoes were microscopically examined immediately after feeding to determine the number of sporozoites in the salivary glands. After 10 days, blood smears from each mouse were microscopically examined to determine the percentage parasitemia. These five mice were then given either the TBV antibody at the desired dose to achieve a 50%, 65% or 85% reduction in oocyst prevalence as required, or no intervention/control (in accordance with the respective treatment arm). A new cohort of 500 naive mosquitoes was then allowed to blood feed on the mice. This mouse-to-mouse transmission cycle was repeated to a maximum of four cycles after the seeding mouse population or until no parasites had been detected in the system for two successive transmission cycles. The PEV and TBV antibodies at each dose (corresponding to a reduction in transmission to mosquitoes of 50%, 65% and 85%) were tested singly and in combination.

Initial parasite density was measured by counting the number of infected red blood cells (out of a total subsample of 1200 erythrocytes). The number of sporozoites in the salivary glands following blood feeding was counted on the logarithmic scale (scores of 0–4 representing 0, 1–10, 11–100, 101–1000, 1000 + sporozoites, respectively). The data are provided in *Table 1—source data 1*.

## Ethics statement

All animal procedures were performed in accordance with the terms of the UK Animals (Scientific Procedures) Act (PPL 70/8788) and were approved by the Imperial College Animal Welfare and Ethical Review Body (AWERB) LASA guidelines were adhered to at all points. The Office of Laboratory Animal Welfare Assurance for Imperial College covers all Public Health Service supported activities involving live vertebrates in the US (no. A5634-01).

## Statistical analysis

The complexity of the population assay requires non-standard methods of statistical analyses that can account for the non-linear dynamics of transmission and stochastic fluctuations seen by the relatively small number of mice used in each generation and treatment arm. These methods need to be able to determine whether the interaction between the different antibodies, that simulate vaccine-triggered antibodies, is below what would be expected if vaccine effects were less strong than expected if effects were multiplied (sub-multiplicative), independent of the presence of the other vaccine (multiplicative) or synergistic, in that effects are enhanced for one or both vaccine types (super-multiplicative).

A density model was developed specifically for this purpose (described in full, [*Sherrard-Smith et al., 2017*], *Figure 1—figure supplement 1*). The structure of the model captures the experimental set up and fits explicitly to parasite densities during successive life stages to generate more precise estimates of simulated vaccine efficacy than the direct comparison of raw data alone, which can fluctuate widely due to chance (as each generation only has five mice) (*Table 1—source data 1*). Stochastic elimination (or resurgence) is possible in the mouse system given the ethical necessity to keep the mouse populations in each generation small; as only five mice are used so transmission could be halted or enhanced by natural variability in the physiological response from each individual mouse. The statistical mode, however, explicitly addresses this variation by modeling the distribution of pathogens at each stage of transmission for each mouse individually. By explicitly incorporating the heterogeneity in the mouse pathogen load the model avoids bias in the inference of the transmission process itself. All parameters were fitted jointly using a Bayesian posterior

distribution in RStan (version 2.13.1, [**Stan Development Team, 2017**]). To ensure robust fits, a non-centered parameterization method was employed (**Papaspiliopoulos et al., 2007**; **Betancourt and Girolami, 2015**). The model parameter fitting was achieved using a Hamiltonian Monte Carlo method (**Stan Development Team, 2017**), warmup was 500 and the subsequent 500 samples from each chain (n = 4) were used for the posterior predictive checks (**Sherrard-Smith et al., 2017**). The model was validated by visualizing the observed raw data measurements of parasite density in mice, the oocyst counts and the logarithmic counts of sporozoites in mosquitoes against model predictions. The data were analyzed at different scales, first taking all data together before breaking down the impact by the dose of the TBV antibody.

The prevalence efficacy against infections in mice is the percentage difference in the proportion of infected hosts between the control and treatment arms of the experiment. The parasite density efficacy against infections in mice is the percentage difference in the mean parasite density per host between the control and treatment arms of the experiment. (Similarly, efficacies can be calculated for the reduction in oocysts or sporozoites in mosquito populations.)

Efficacy estimates were generated for each arm of the experiment, for each posterior draw of the model (2000 posterior draws). This allows mean and 95% credible intervals to be calculated for each treatment group (c) and across mosquito biting rates (m) and transmission cycles (i). Let $P_{c,m,i}^{j}$ indicate the prevalence of infected mice (j = 1) or the mean asexual parasite density in the mouse population (j = 2). Treatment arm 0 represents the control data, and c indicates treatments 1 to 7 (TBV antibody at 50%, 65%, 85% dose singly, PEV antibody singly, PEV and TBV antibody at 50%, 65% and 85% dose together), such that,

$$E_{c,m,i}^{j} = \frac{P_{0,m,i}^{j} - P_{c,m,i}^{j}}{P_{0,m,i}^{j}} \times 100 \tag{1}$$

where $E_{c,m,i}^{j}$ is either the parasite prevalence efficacy (j = 1) or density efficacy (j = 2) against infections in mice as estimated by the posterior predictions of the density model (**Table 1**).

To statistically assess whether the interaction between PEV and TBV antibodies are antagonistic, independent or synergistic, efficacy estimates against infections in mice for combined antibody treatments were compared to the expected estimates if antibodies had an independent impact (expected efficacy). To estimate this expected efficacy the single antibody treatment groups were combined, as follows (**VanderWeele and Knol, 2014**):

$$Expected\,Efficacy = E_{pev} * \left(1 - E_{tbv,d}\right) + E_{tbv,d} \tag{2}$$

where $E_{tbv,d}$ is the prevalence or density efficacy for the TBV antibody treatment alone (at the specified TBV dose of d = 50%, 65% or 85%), and $E_{pev}$ is the prevalence or density efficacy for the PEV alone. The ratio between efficacies for the combined treatments and the expected efficacy for matched treatments was used to assess synergy. A synergistic interaction is indicated when this ratio is greater than 1, an independent interaction when equal to one and an antagonistic impact if less than 1 (**Figure 4**). The 95% credible intervals were calculated to give statistical support. Statistical evidence of a difference in treatment and control experiments (p-value) was defined as one minus the proportion of iterations from the model simulations that were greater than 1.

The observed synergy can be explained because the presence of TBV antibody reduces the sporozoite score (as a measure of parasite density in the mosquito population) which allows the PEV antibodies to achieve a greater efficiency. To understand whether the combined TBV antibody treatments improved the action of the PEV at higher parasite densities, the raw data recording the sporozoite scores of each mouse in the single treatment TBV antibody groups were plotted for each dose and across transmission cycles to demonstrate that there are progressively more mosquitoes without infection and progressively fewer mosquitoes with heavy infections (**Figure 3**). This highlights that the PEV in the combined treatment groups is acting against progressively fewer parasites in each transmission cycle because of the initial action of the TBV antibody. Simple analysis of variance was performed to confirm that the presence of the respective antibody type (PEV acting against infections in mice, TBV acting against sporozoites in mosquitoes), as a binary covariate, could explain reduced parasite counts in mosquitoes or mice. To demonstrate that the PEV antibody has a density-dependent impact, binomial logistic regression curves were fitted to determine the relationship

between the prevalence efficacy of the PEV antibody and the parasite density in mosquitoes (measured as mean sporozoite score for each mosquito biting rate and transmission cycle, n = 16). Parameters describing the regression curves were fitted using a Bayesian posterior distribution in RStan (version 2.13.1, [*Stan Development Team, 2017*]). The model parameter fitting was achieved using a Hamiltonian Monte Carlo method (*Stan Development Team, 2017*), warmup was 1000 and the subsequent 1000 samples from each chain (n = 4) were used for the posterior predictive checks (*Stan Development Team, 2017*). All data analysis were conducted using the statistical software R (version 3.2.2; [*Core Team, 2014*]).

## Acknowledgements

The data used in this manuscript are listed in Supplementary information *Table 1—source data 1*. We thank Mark Tunnicliff for mosquito production. AMB, TSC, AG, ESS and MJM formulated the theory and prediction for the concepts presented. AMB, TSC and KAS contributed to experimental conception and design. Acquisition, analysis and/or interpretation of data were completed by ESS, MB, AMB, KAS, LMU and FA. ESS, TSC, AG and AMB drafted the article and all authors revised it critically for important intellectual content. The authors acknowledge PATH's Malaria Vaccine Initiative for funding this work. TSC thank the UK Medical Research Council (MRC)/UK Department for International Development (DFID) under the MRC/DFID Concordat agreement. AMB thanks the MRC (grant number MR/N00227X/1) for funding. MB is supported under EPSRC grant EP/J016934/1.

## Additional information

### Funding

| Funder | Grant reference number | Author |
|---|---|---|
| PATH | GAT.0888-11-06546-COL | Azra C Ghani<br>Thomas S Churcher<br>Andrew M Blagborough |
| Medical Research Council | MR/N00227X/1 | Andrew M Blagborough |
| Engineering and Physical Sciences Research Council | EP/J016934/1 | Michael Betancourt |

The funders had no role in study design, data collection and interpretation, or the decision to submit the work for publication.

### Author contributions

Ellie Sherrard-Smith, Formal analysis, Validation, Investigation, Visualization, Methodology, Writing—original draft, Writing—review and editing; Katarzyna A Sala, Data curation, Investigation, Methodology, Writing—review and editing; Michael Betancourt, Formal analysis, Validation, Methodology, Writing—review and editing; Leanna M Upton, Fiona Angrisano, Data curation, Methodology, Writing—review and editing; Merribeth J Morin, Conceptualization, Supervision, Project administration, Writing—review and editing; Azra C Ghani, Thomas S Churcher, Conceptualization, Supervision, Funding acquisition, Methodology, Writing—original draft, Project administration, Writing—review and editing; Andrew M Blagborough, Conceptualization, Data curation, Supervision, Funding acquisition, Methodology, Writing—original draft, Writing—review and editing

### Author ORCIDs

Ellie Sherrard-Smith [iD] http://orcid.org/0000-0001-8317-7992
Michael Betancourt [iD] http://orcid.org/0000-0002-2900-0931
Fiona Angrisano [iD] http://orcid.org/0000-0002-0457-5982
Thomas S Churcher [iD] http://orcid.org/0000-0002-8442-0525

## Ethics

Animal experimentation: All animal procedures were performed in accordance with the terms of the UK Animals (Scientific Procedures) Act (PPL 70/8788) and were approved by the Imperial College Animal Welfare and Ethical Review Body (AWERB) LASA guidelines were adhered to at all points. The Office of Laboratory Animal Welfare Assurance for Imperial College covers all Public Health Service supported activities involving live vertebrates in the US (no. A5634-01).

## Decision letter and Author response

Decision letter https://doi.org/10.7554/eLife.35213.012
Author response https://doi.org/10.7554/eLife.35213.013

## Additional files

### Supplementary files

• Transparent reporting form
DOI: https://doi.org/10.7554/eLife.35213.010

### Data availability

All data generated or analysed during this study are included in the manuscript and supporting files.

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
