## [Decision Letter]

Thank you for submitting your article "Synergy in anti-malarial pre-erythrocytic and transmission-blocking antibodies is achieved by reducing parasite density" for consideration by *eLife*. Your article has been reviewed by three peer reviewers, and the evaluation has been overseen by a Reviewing Editor and Wendy Garrett as the Senior Editor. The reviewers have opted to remain anonymous.

The reviewers have discussed the reviews with one another and the Reviewing Editor has drafted this decision to help you prepare a revised submission.

Summary:

The manuscript by Sherrard-Smith et al. addresses an important question in malaria vaccine development, namely the impact of multistage vaccination approaches, e.g., pre-erythrocytic (PEV) and transmission blocking (TBV), in various transmission settings on the overall malaria infection. The paper is commendable for being cross-disciplinary (i.e. mouse experiments and modelling) and for its clarity of presentation. Figure 1 neatly provides an overview of the raw data then Figure 2 with contains the essence of the modelling output. The observation that PEV + TBV is synergistic, and specifically through reducing sporozoite counts thus giving imperfect PEVs a better chance to work is well supported for this model set up. This conclusion has been inferred from theoretical models before, and this is the first demonstration using laboratory data. The manuscript is well written, and the study design clearly laid out and well executed.

Essential revisions:

There are a few issues that the authors need to address before the manuscript is considered for an acceptance in *eLife*.

1) Please consider performing additional experiments that would also include varying PEV efficacy. Should you consider that such experiments would not be interesting, please justify your reason(s).

2) The other revisions concern expansions or explanations of the following points:

a) Concerning the relevance of synergy, it would be assumed that a highly efficacious PEV or TBV will not require synergy and it is only the 50% TBV where the effect is really noticeable.

b) Expand on the applicability of the murine population model to human vaccine trials and particularly how would the variabilities in human transmission be built into this assay or modeled from this populations assay.

c) Considering situations that commonly exists in the field and that do impact transmission, e.g., vaccine non-responders or a super-transmitter owing to a high oocyct counts, comment how such situations would impact the model, especially since previous preclinical work has not translated to human results.

d) Please explain the selection of the three different efficacies for TBV alone. Comment on the prevalence and the oocyst counts in the control group and the variations from experiment to experiments.

e) Finally, how could polymorphisms of TBV and PEV antigen vaccine candidates encountered in the field be accounted for in the murine population assay and how that may impact your results.

---

## [Author Response]

Essential revisions:

*There are a few issues that the authors need to address before the manuscript is considered for an acceptance in* eLife.

1) Please consider performing additional experiments that would also include varying PEV efficacy. Should you consider that such experiments would not be interesting, please justify your reason(s).

We agree with the reviewers that having multiple TBV doses would make the manuscript more symmetrical and exhaustive. Having seriously considered the inclusion of alternative PEV doses we feel that these experiments are beyond the scope of this piece of work for a range of reasons.

Firstly, the series of experiments described here were designed to examine the potential synergy in anti-malarial pre-erythrocytic and transmission-blocking antibodies. The PEV dose was chosen to approximately match the efficacy of RTS,S, the only malaria vaccine to have been tested in a Phase III trial. The work was intended to provide the evidence-base for why development of a partially effective TBV would be beneficial for malaria control. This was necessary as a TBV provides no direct protection to the recipient and must rely solely on community protection, something that is complex and very challenging to measure in controlled human settings. It is hoped that future generation PEVs might have a higher efficacy, the rational for investing in this additional efficacy is self-evident because the more efficacious the vaccine the more personal protection it provides. The main reason for attempting this work is that the indirect nature of how effective TBVs might be if combined with PEVs means that the outcome of using both vaccine is less obvious and potentially less predictable.

Secondly, although evidently of interest, we consider that other examinations of PEV efficacy could be more relevant and timely to public health settings than a relatively simple examination of a lower or higher static PEV efficacy. RTS,S-induced antibody concentration (and efficacy) is known to decay relatively quickly, and coverage is expected to be lower than the 100% examined in this manuscript. In the future, rather than a simple examination of a lower/higher than 50% PEV efficacy, we are interested in looking at the impact of “decaying” vaccine efficacy over time, and in examining multiple vaccine coverages (i.e. number of individuals immunized). We strongly see this as a distinct piece of work rather than an addition to the current manuscript. Additionally, we consider the experiments presented here to be exceptionally timely considering the current status of the vaccine development pipeline, and the observed findings from current efficacy trials in the field.

Finally, financially and logistically conducting these experiments is not a simple matter and is not immediately feasible. Each experiment takes approximately 165 days and requires the use of 125 mice. The addition of an additional PEV dose would require a minimum of 2 additional separate population experiments (PEV 65% vs PEV 65% + TBV 50%). An exhaustive examination of all doses (3 TBV + 3 PEV) would require an additional 7 experiments equating to over 875 mice. Any additional experiments would require further funding and time that is beyond our capacity at present. The ethical necessity for the experiments would also require some consideration given the points below.

2) The other revisions concern expansions or explanations of the following points:a) Concerning the relevance of synergy, it would be assumed that a highly efficacious PEV or TBV will not require synergy and it is only the 50% TBV where the effect is really noticeable.

The reviewers are correct in stating that the 85% TBV (alone) treatment performed to the same level as the combination dose 85% TBV + 50% PEV, indicating that in this situation the high dose TBV was sufficient to eliminate the parasite. In this setting there was no necessity for the PEV or synergy to eliminate the parasite because the 85% achieves this alone. Nevertheless, this is unlikely to be the case in high transmission field settings as population size will be larger, vaccine coverage will be less than 100%, antibody concentration will likely decay relatively quickly, and human migration will mean cases are imported from outside the vaccinated area. Transmission dynamics mathematical models indicate that a single highly efficacious vaccine alone will be insufficient to eliminate the parasite in all settings, so control will always benefit from any synergistic interaction. We hypothesise that synergism will still be happening in 85% TBV + 50% PEV group and that, on average, the parasite would have been eliminated faster than in the 85% TBV group alone. The population assay is not sensitive enough to detect this subtle difference in control though it may be relatively important in real life-settings and could make the difference between parasite elimination and disease persistence. We have added the following to the Discussion section.

“The greatest synergy was observed in the lower TBV dose group though it is likely to operate across all TBV and PEV doses when vaccine reduces parasite density but fails to clear infection from host or vector. The 85% TBV dose administered alone eliminated the parasite over a single generation without the action of the PEV so there was no opportunity to show synergy. A 100% efficacious PEV or TBV would not require augmenting with an alternative vaccine though their efficacy is still likely to drop over time as antibodies decay so combining highly effective vaccine may still be advantageous depending on the relative rates of antibody loss.”

b) Expand on the applicability of the murine population model to human vaccine trials and particularly how would the variabilities in human transmission be built into this assay or modeled from this populations assay.

The experiment was designed to capture a range of transmission intensities simulated by using multiple mosquito bites per mouse (1, 2, 5 and 10 bites per mouse). The statistical approach used here was designed specifically to encompass this experimental design, account for high parasite over-dispersion and allow the impact of vaccine-induced antibodies on parasite density to be assessed. The population assay design meant that transmission intensity varied between groups of mice not within the same group. Transmission dynamics of malaria in the field suggest substantial within population heterogeneity in transmission as some people are bitten substantially more than others. Within population heterogeneity could be included within the mouse population assay to further investigate the impact in a control laboratory setting.

It is currently unclear how a TBV would be evaluated in humans under natural conditions. It would be possible to repeat the experimental design but using groups of humans instead of groups of mice though the feasibility and the ethics of the experiment would need to be thoroughly assessed. To investigate the synergy regulated by parasite density hypothesis within a standard human Phase III RCT would require multiple repeat measures of the density of infections in mosquitoes and people monitored during the trial. The work presented here could support the design of appropriately powered RCTs to fully assess the impact of combining vaccine components with alternative mechanisms of action.

Whilst the murine system described here uses *P. berghei,* the PEV component used targets CSP, the key antigen within RTS,S. Thus, the mechanism of action of the PEV antibodies administered *i.v* is matched to the antibody component of the RTS,S vaccine. For the TBV component, the *P. berghei* strainused here is genetically modified to express the human TBV candidate, *P. falciparum* P25 (Pfs25) in place of its *P. berghei* counterpart.Thus, a proven anti-*falciparum* TBV mAb (4B7) can be used directly within this model. This gives enhanced confidence that the synergistic impact measured would be reflected in human data (were it available). Of course, mice are not human and *P. berghei* is not *P. falciparum*. The use of model systems to examine malarial transmission is clearly necessary within multiple circumstances. The value of these studies is undoubtedly to examine the relationship between multiple anti-malarial antibody moieties, parasitic density, and ability to achieve elimination in a system that is tractable, and where these experiments are technically and ethically possible. Following the establishment of these basic dynamics in model systems, these findings should be used to inform studies with human malarial parasites in the future.

We have updated the Discussion section:

“Whilst the murine system uses *P. berghei*, the mechanism of action of the PEV antibodies administered is matched to the antibody-based mechanism of the RTS,S vaccine, i.e. sporozoite invasion of the liver is inhibited by the presence of CSP-targeted antibodies in mice both in vivo and ex vivo (Grüner et al., 2003). The *P. berghei* strain used here is genetically modified to express the human TBV candidate, *P. falciparum* P25 (Pfs25), in place of its *P. berghei* counterpart. Thus, a proven anti-*falciparum* TBV mAb (4B7) can be used directly within this model.*”*

In addition, we have added the following to the Discussion section (also in response to the question below).

“The epidemiological importance of any synergistic interaction between vaccine types in the field is hard to predict and will depend on many confounding factors such as human immunity, drug treatment, vector susceptibility, antigen escape, amongst others. Transmission is likely to be highly heterogeneous, caused by factors such as vaccine non-responders and super-transmitting hosts. The impact of different types of heterogeneity can be investigated under controlled laboratory scenarios using the murine population assay, varying the vaccinated coverage, antibody dose and changing biting heterogeneity within a population. This could help understand the relative importance of these different heterogeneities and could be used to support the design of appropriately powered Phase III trials (or alternative trial designs) to fully assess the impact of combining vaccine components with alternative mechanisms of action.”

c) Considering situations that commonly exists in the field and that do impact transmission, e.g., vaccine non-responders or a super-transmitter owing to a high oocyct counts, comment how such situations would impact the model, especially since previous preclinical work has not translated to human results.

We agree that there will be greater uncertainty in field trials because so many (potentially undetectable and unmeasurable) covariates are impacting on transmission. Nevertheless, we would expect to observe some synergy from partially effective vaccines given the murine experimental results. We would argue that the population assay provides a powerful tool to help explain why epidemiological characteristics (such as heterogeneity in transmission) might influence intervention effectiveness and these results might feed into why pre-clinical work might not translate into human results as highlighted by the reviewer. In light of this we are currently investigating incomplete coverage of hosts using this model. This is posited to be equivalent to non-responders or less than 100% vaccine coverage.

We have added the following to the Discussion section:

“The epidemiological importance of any synergistic interaction between vaccine types in the field is hard to predict and will depend on many confounding factors such as human immunity, drug treatment, vector susceptibility, antigen escape, amongst others. Transmission is likely to be highly heterogeneous caused by factors such as vaccine non-responders and super-transmitting hosts. The impact of different types of heterogeneity can be investigated under controlled laboratory scenarios using the murine population assay, varying the vaccinated coverage, antibody dose and changing biting heterogeneity with a population. This could help understand the relative importance of these different heterogeneities and could be used to support the design of appropriately powered Phase III trials (or alternative trial designs) to fully assess the impact of combining vaccine components with alternative mechanisms of action.”

d) Please explain the selection of the three different efficacies for TBV alone. Comment on the prevalence and the oocyst counts in the control group and the variations from experiment to experiments.

In terms of PEVs, the most advanced malaria vaccine achieved just a 36.3% efficacy in early trials (and at the time of designing the present work). There has been much debate over the approval of this vaccine. Given this, the next malaria vaccine to be developed may well need to perform better. We considered 50% as a lower limit. Similarly, producing a highly effective (TBV) vaccine will be undoubtedly challenging in reality so we capped the upper limit to 85% within this study. The lowest TBV efficacy of 50% was chosen under the premise that a TBV of <50% reduction in oocyst prevalence would not practically be considered for release as part of a vaccine development pipeline. 65% was chosen as an intermediate efficacy point. In term of oocyst counts in control groups and experimental variation while titrating doses of 4B7, a series of transfused mAb-4B7 doses were tested in multiple (n = 6) direct feeding assays, each with an individual control. The statistical analysis used here is expressly designed/chosen to examine experimental variation in non-parametric oocyst counts over multiple experiments over a range of transmission settings (see Churcher et al., 2012, Blagborough et al., 2013, Upton et al., 2015, Churcher at al., 2017 and Sherrard-Smith et al., 2017, for more information).

We have added in subsection “Transmission-blocking vaccine surrogate: Monoclonal antibody 4B7 (mAb-4B7). The TBV”:

“Given the severity of malaria, the World Health Organization (WHO), the Strategic Advisory Group of Experts (SAGE) on Immunization and the Malaria Policy Advisory Committee (MPAC) recommended the RTS,S vaccine could be implemented in pilot countries in October 2015 (http://www.malariavaccine.org/malaria-and-vaccines/first-generation-vaccine/rtss, accessed 04/04/2018) when the vaccine was demonstrating relatively low efficacies of just 36.3% in children aged 5 to 17 months (RTSS Clinical Trials Partnership 2015). These TBV doses were chosen to bridge a range of malaria vaccine efficacies that might be acceptable by WHO, SAGE and MPAC.”

e) Finally, how could polymorphisms of TBV and PEV antigen vaccine candidates encountered in the field be accounted for in the murine population assay and how that may impact your results.

When considering the overall practical impact of all antimalarial vaccines, and, more broadly, all interventions, (bloodstage, pre-erythrocytic and transmission blocking) we fully agree that the impact of polymorphisms and the associated possibility of pathogen escape must be considered. The interventions (both PEV and TBV) used here are monoclonal antibodies against both CSP and P25 respectively. As such, the interventions utilized target only a very limited number of epitopes with pre-proven efficacy. Therefore, the experimental design here, although well suited to examine parasitic density, is not optimised for the emergence of single or multiple polymorphisms over time. When considering the use of vaccines in the field, it is likely that the introduction of polymorphisms will reduce overall vaccine efficacy/effectiveness. In the future, complimentary studies that examine the potential impact of this could be designed using this system, and administration of vaccines with a broader epitope range, followed by sequential transmission of parasites from mouse-to-mosquito-to-mouse. Specific polymorphism could be introduced by transgenesis, or screening for “naturally” occurring spontaneous polymorphism could be performed, in parallel with constant monitoring of efficacy over time. However, this is outside the remit of the current study.

We have highlighted the possibility of antigen escape in the Discussion section.

“The epidemiological importance of any synergistic interaction between vaccine types in the field is hard to predict and will depend on many confounding factors such as human immunity, drug treatment, vector susceptibility, antigen escape, amongst others.”